# Aerodynamic Effects of Ceiling and Ground Vicinity on Flapping Wings

Xueguang Meng [1], Yinghui Han [1], Zengshuang Chen [1], Anas Ghaffar [1] and Gang Chen [1,2,*]

1   State Key Laboratory for Strength and Vibration of Mechanical Structures, School of Aerospace,
    Xi'an Jiaotong University, Xi'an 710049, China; mengxg@xjtu.edu.cn (X.M.); hyh2020@stu.xjtu.edu.cn (Y.H.);
    chenzengshuang@stu.xjtu.edu.cn (Z.C.); anaxgh1@gmail.com (A.G.)
2   Shaanxi Key Laboratory of Environment and Control for Flight Vehicle, School of Aerospace,
    Xi'an Jiaotong University, Xi'an 710049, China
*   Correspondence: aachengang@xjtu.edu.cn

**Abstract:** The combined ceiling and ground effect on the aerodynamics of a hovering flapping wing is investigated using numerical simulations. In the simulations, the wing was located between the ceiling and the ground. Simulations were carried out for different wall clearances at two Reynolds numbers ($Re$ = 10 and 100). Special efforts were paid to whether there exists aerodynamic coupling between the ceiling effect and the ground effect. At $Re$ = 10, the combined ceiling and ground effect increases the aerodynamic forces monotonically through two effects, namely the narrow-channel effect and the downwash-reducing effect. Additionally, there exists a coupling effect of the ceiling and the ground for the combined case at $Re$ = 10, where the force enhancement of the combined effect is much more significant than the sum of the ceiling-only effect and the ground-only effect. At $Re$ = 100, the combined effect of ceiling and ground causes three non-monotonic force regimes (force enhancement, reduction and recovery) with increasing wall clearance. The narrow-channel effect at $Re$ = 100 leads to a monotonic force trend, while the downwash-reducing effect results in a non-monotonic force trend. The two effects eventually lead to the three force regimes at $Re$ = 100. Unlike the $Re$ = 10 case, the coupling effect at $Re$ = 100 is small.

**Keywords:** ceiling effect; ground effect; coupling effect; insect flight; aerodynamics; flapping wing; numerical simulation

---





## 1. Introduction

Micro Air Vehicles (MAVs), with small size as one of their key features, are attracting more and more attention from all over the world. One of the biggest advantages for the MAVs is their ability to carry out missions in a confined space, such as flying in the jungle to perform military reconnaissance, or searching for survivors in the ruins after a great earthquake. However, in environments that are so crowded and full of obstacles, it is inevitable that these MAVs will fly near a substrate, experiencing a wall effect. A similar scenario in the daily flight of small birds and tiny insects—which use flapping wings as their source of lift—is flight in bushes full of leaves and branches [1]. These situations will also project a wall effect on the aerodynamics of flapping wings. Finding out the impact a wall will have on the aerodynamics of flapping wings is essential to understanding the behaviors of near-wall flight in animals, as well as the development of flapping MAVs.

Before investigating the wall effect, much effort was devoted to exploring the underlying mechanisms of the generation of high lift with flapping wings [2–4], based on high-speed-photography-recorded wing kinematics. The revealed unsteady aerodynamic mechanisms include delayed stall, rapid acceleration (or added mass), rapid pitching rotation, clap-and-fling, rowing, etc. [5–13]. While rapid acceleration and clap-and-fling are only used by specific insects for performing particular movements, the delayed stall

mechanism is used by most insects. It is related to the continued attachment of the leading-edge vortex (LEV) to the wing in individual strokes, and is the leading cause of the high lift coefficient of the insect's flapping wing. In addition to the attached LEV, the tip vortex (TV), trailing-edge vortex (TEV) and root vortex (RV) continuously detach from the wing, forming a vortex wake. The attached LEV, together with the detached vortices forms a doughnut-shaped vortex ring that also produces a downwash [14]. Since there is significant fluid separation above the wing and a complex vortex wake under the wing, it is easy to predict that a surface above or below the wing will affect the vortex structures around the wing and the induced downwash of the vortex ring, thus leading to changes in the aerodynamics of the flapping wing.

The ground effect, when the wall is below the flapping wing, has already been investigated. It is inspired by the long-existing study of the ground effect on a fixed aircraft wing. For a fixed-symmetry airfoil at a positive angle of attack, the ground effect generally causes increase in lift and a larger lift-to-drag ratio by increasing the pressure on the lower surface [15,16]. However, for different airfoil configurations, the aerodynamic effects caused by the ground are different. For an asymmetric airfoil in Ref. [17], the ground decreased the lift at a negative angle of attack and caused a non-monotonic lift trend at a low-to-moderate angle of attack. By installing a front wing, a racing car can also take advantage of the ground effect to increase the downward force, and this ground effect can be further enhanced by adding a gurney flap on the fixed wing [18,19]. It is interesting to know whether the ground effect for a flapping wing is the same as that for a fixed wing. The pioneering work of Lu et al. numerically studied the ground effect on a two-dimensional (2D) flapping foil at a Reynolds number of 100. Unlike those for the fixed wing, the aerodynamics of the flapping wing were found to experience three non-monotonic force regimes as the wall clearance increased [20]. This non-monotonic force trend is further confirmed by later studies, including a particle image velocimetry (PIV) work [21] on a 2-D hovering airfoil at $Re = 1000$, and a numerical simulation [22] using a NACA-0015 airfoil at $Re = 100$. In the PIV study, the force increment was mainly due to a stronger wake capture, the force decrease was due to a weak LEV and the negative effect of the wake, and the recovery was because of the LEV's attachment to the airfoil during mid-stroke. In more recent research, the ground effect on a three-dimensional (3D) flapping insect wing model was studied at $Re = 100$ and 5000. Both numerical and experimental studies were carried out in this work, and they concluded that the effect on a 3D wing is similar, but less noteworthy, compared to a 2D wing. It was reasoned that the "three force regimes" effect could be attributed to the positional shift in the vortex wake, causing a non-monotonic trend in the downwash [23].

The work of a recent numerical 3D study extends the $Re$ to as low as 10 and compares the ground effect at low and high $Re$ [24]. Unlike the above-mentioned "three force regimes" at $Re = 100$ or higher, it revealed that reducing the $Re$ to a very low level causes a monotonic "single force regime". The force increases monotonically with reducing ground clearance. The different force trend behaviors stem from different evolution processes of the vortex wake. Additionally, the "cramming effect" also contributes to the force enhancement by increasing the positive pressure on the wing surface.

Studies also show the aerodynamic ground effect on the wing in an insect's take-off [25–28]. The experimental study on the takeoff of a 3D beetle model wing at $Re = 10,000$ showed that the overall vertical force for the initial two strokes was increased by the LEV enhancement of the downstroke [25]. However, other works argue that the ground effect on the oscillating wings of a drone fly and a fruit fly was insignificant during takeoff [26–28]. It is believed that the rapid dynamic process of takeoff is responsible for the negligible effect. This is because the insect wings rapidly move off the ground to above a wing-chord-length distance of 3 within one or two strokes [24].

The ceiling effect, when the wall is above the flapping wing, was not investigated until recently, and was inspired by the observed landing flight of a bee on a ceiling (inverted surface) and a flapping-wing MAV that could take off and perch on the overhangs of materials such as a natural leaf, wood, glass, etc. [29,30]. Studies claim that the aerodynamic forces

on a hovering insect wing increase monotonically as the wing approaches the ceiling [31]. They state that the ceiling's presence acts like a mirror, as if a mirroring LEV exists there, thus increasing the relative oncoming flow velocity to the wing, and also producing an upwash to be experienced by the wing. Another work further studied the ceiling effect in forward flight, and comprehended that the aerodynamic forces monotonically increase with reducing ceiling clearance at a lower advance ratio ($J$), but that the ceiling effect reduces with increasing $J$ [32].

As an initial stage, the aforementioned works [20–32] studied the ground effect and the ceiling effect on flapping wings separately. However, an actual flapping insect often flies in confined areas where the ceiling and the ground co-exist in the flapping wing's vicinity. This situation leads to two problems that need to be clarified. The first answer to obtain is the combined ceiling and ground effect on the aerodynamics of the flapping wing. Second, it is necessary to determine whether there exists aerodynamic coupling between the ceiling effect and the ground effect. If no coupling exists, then the combined ceiling and ground effect on a flapping wing is merely a superposition of the separate ceiling and ground effect. However, if some coupling exists, then the aerodynamic effect will differ from the superposition of the separate ceiling and ground effects. In this study, we will try to answer the above two queries. This paper is organized as follows: First, the wing motion and numerical method are introduced. Second, the combined effect of the ceiling and ground on the aerodynamic forces at $Re = 10$ is investigated. Then, the reasons for the combined effect as well as the coupling effect at $Re = 10$ are answered. Following this, the same situations at $Re = 100$ are investigated. Finally, conclusions are drawn based on the findings of this work.

## 2. Materials and Methods

### 2.1. Wing Motion

As shown in Figure 1a, two three-dimensional models of the insect wing were made to flap horizontally in the middle of two horizontal surfaces: a ceiling above and a ground below the wings. For better visualization, the insect's body is represented in Figure 1a, but was not considered in actual computations. The wing's planform shape resembled a hoverfly wing, as shown in Figure 1b. The wing's aspect ratio $R/c$ was 3.75 ($R$ and $c$ being the wing length and mean chord length, respectively) [33], and $R$ was equal to 13.20 mm for the actual hoverfly wing. The radius of gyration of the wing ($r_2$) was $0.56R$. We modeled wing as a flat plate with rounded edges, where its thickness was 3% of $c_l$ (local chord length). In Figure 1c, $D_c$ and $D_g$ are used to denote the distance between the wing base and the wall (ceiling and ground, respectively) and named as the wall clearances. In this paper, $D_c$ equals $D_g$, denoted by $D$.

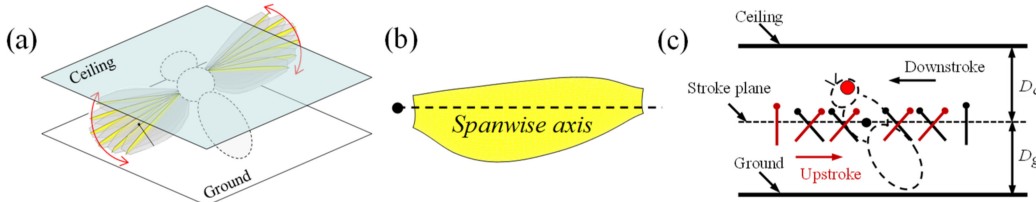

**Figure 1.** (**a**) Schematic motion of the wing between the ceiling and the ground, a wing model is shown as an insect; (**b**) wing's planform shape; and (**c**) side view of a wing section's motion. The black dot indicates the wing base, and $D_c$ and $D_g$ mean the ceiling clearance and the ground clearance, respectively. In this paper, $D_c = D_g = D$. The position of the section on the wing is shown in (**a**).

The kinematics of the wings implemented in our study is referred to as "normal hovering" which is a simplified model of the actual insect wing kinematics with a horizontal stroke plane. The spanwise axis of the wing, which connects the wing base to the wing tip, rotates about the wing base in the stroke plane reciprocally, and at the same time, the wing flips around the spanwise axis (see Figure 2a). Following the conventional naming

methods, the forward and backward motions are called the downstroke and the upstroke, respectively. In Figure 2a, two independent angles, $\phi(t)$ and $\psi(t)$, are defined to describe the wing motion relative to the wing base. $\phi(t)$ means the stroke position angle, and $\psi(t)$ means the rotational angle. In the inertial frame $OXYZ$, the origin $O$ coincides with the wing base and the $XOY$ plane overlaps with the stroke plane. $\phi(t)$ is the angle between the spanwise axis and Y-axis (Figure 2a). Time variation of $\phi(t)$ is given by:

$$\phi(t) = 0.5\,\Phi\cos(2\pi nt) \tag{1}$$

where $\Phi$ is called the stroke amplitude, and $n$ represents the wingbeat frequency and equals about 150 Hz.

The rotation angle $\psi(t)$ is the angle between the wing surface plane and the stroke plane. The wing moves with a constant angle $\psi$ during the mid-portion of each half stroke while it flips over at stroke reversals. $\alpha$ denotes the mid-stroke angles of incidence. In the downstroke, $\alpha = \psi$, while in the upstroke, $\alpha = 180^\circ - \psi$. The function of $\psi(t)$ is described in our previous work [31,32].

Figure 2b demonstrates the time course curves of the two angles ($\psi(t)$ and $\phi(t)$) over a stroke period. The dimensionless time $\tau$ is used, which equals $t/T - (n - 1)$ at the $n$th beat to implement the period normalization, and facilitates the representation of the different stages of any beat ($T$ is the wingbeat period, equal to $1/n$). Thus, $\tau$ ranges from 0 to 1 for any complete flapping cycle. The first half is the downstroke ($0 < \tau < 0.5$) and the second half is the upstroke ($0.5 < \tau < 1$). As per the data measured [33–35], $\Phi$ and $\alpha$ are fixed to be 110 and 40 degrees, respectively. The wing rotation interval for a stroke's reversal is set as 30% of $T$.

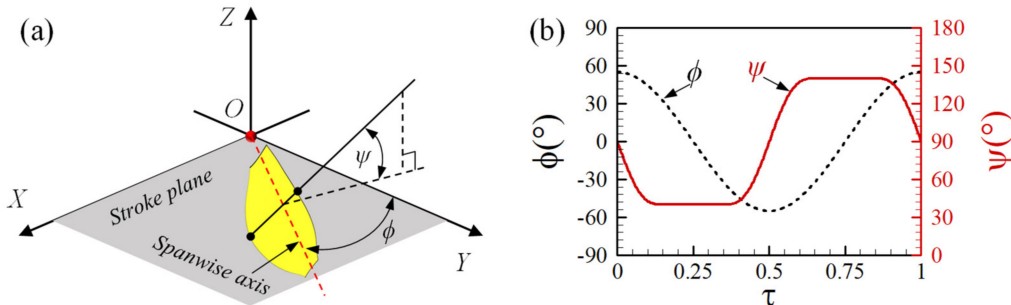

**Figure 2.** (**a**) Coordinate system and two angles ($\psi$ and $\phi$) of the flapping motion; (**b**) time course curves of $\psi(\tau)$ and $\phi(\tau)$ in a flapping cycle.

### 2.2. Numerical Methods

Since the Reynolds number and the velocity of the flapping wing are very low, the laminar incompressible 3D Navier–Stokes equations are the flow's governing equations. The dimensionless form is given by:

$$\nabla \cdot \boldsymbol{u} = 0 \tag{2}$$

$$\frac{\partial \boldsymbol{u}}{\partial t^*} + \boldsymbol{u} \cdot \nabla \boldsymbol{u} = -\nabla\,p^* + \frac{1}{Re}\nabla^2\boldsymbol{u} \tag{3}$$

For the above equations, the non-dimensional parameters of fluid velocity, pressure, and time are abbreviated as $\boldsymbol{u}$, $p^*$ and $t^*$, respectively. The wing's mean velocity at the radius of gyration $U = 2\Phi nr_2$, the mean chord length $c$, and the time $c/U$ were employed for nondimensionalization. $Re$ is the Reynolds number, calculated as $Re = cU/\upsilon$ and $\upsilon$ means the fluid kinematics viscosity.

The N-S equations were numerically solved for moving overset meshes using the method of artificial compressibility, as in previous works [31,36]. In the following, we just provide the method's outline. First, a general time-dependent coordinate transformation

was used to transform the equations from the inertial Cartesian coordinate system to the body-fixed and non-inertial curvilinear coordinate system. A three-point backwards differencing scheme of second-order accuracy was implemented to discretize the time derivatives of the momentum equation. To solve the time discretized moment equations for a divergence-free velocity at a new physical time level, a pseudo-time level was introduced into the equations. Then a pseudo-time derivative of pressure divided by the artificial compressibility constant was added in the continuity equations. The resultant hyperbolic equations system was iterated using pseudo-time until the pressure derivative became zero in pseudo-time; therefore, the velocity divergence also became zero for the new time instance. The central difference method was employed for approximating the viscous-flux derivative in the equations. The upwind differencing formula based on the flux-difference-splitting method was employed for convective flux derivatives. The third-order and second-order upwind differencing methods were employed for interior points and boundary points, respectively. The overall accuracy of the code was two-order. The algorithm's details are present in Refs. [37,38]. The following are the boundary conditions in detail. At far-field boundaries, there was no way to simply specify which far-field boundary is the inflow or outflow boundary, because the flapping wings were in the hovering state. Therefore, at each far-field boundary, the calculation result of the previous time step was used to determine whether the boundary adopted the inflow boundary condition or the outflow boundary condition. If it adopted an inflow boundary condition, the velocity components would be set to zero because of the hovering condition and flight speed equal to zero, and pressure was extrapolated from the interior. Meanwhile, if it adopted an outflow boundary condition, pressure would be set equal to the air static pressure, and velocity was extrapolated from the interior. At the wing surface, the no-slip boundary and impermeable wall conditions were applied, and the pressure was calculated through the normal component of the momentum equation, written in the moving coordinate system. On the ceiling and the ground, the pressure gradient normal to the wall and the velocity were fixed as zero.

The domain discretizing grid had two curvilinear (body-fitted) wing grids extending from the surface of the wing to various distances. These wing grids were within another background Cartesian grid covering the domain until the far fields (Figure 3). The moving wing grids captured detailed aerodynamics, e.g., boundary layers, vortices, etc. The stationary background mesh calculated the solution until the far field, capturing the flow structures such as the vortex wake. Data were interpolated from one grid to the other using a tri-linear interpolation at inter-grid boundary points. A Poisson solver was used to generate wing grids from Hilgenstock's work [39], with the background grid being algebraically generated.

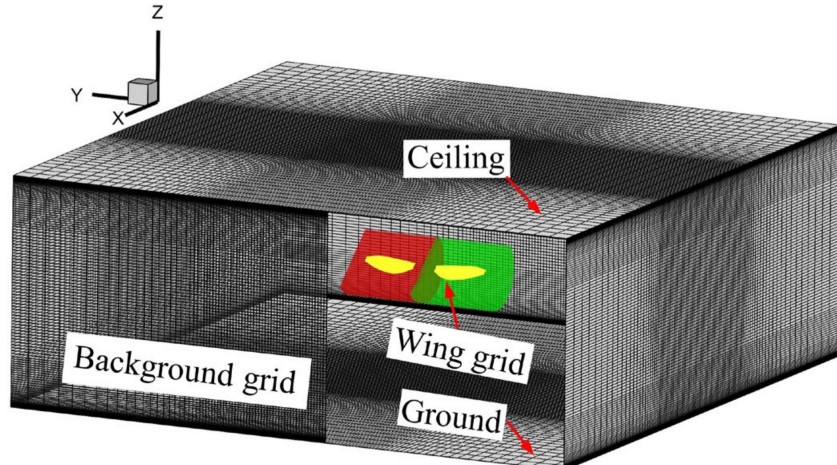

**Figure 3.** Wing meshes in a background mesh.

The wing grid's dimensions for this work were $54 \times 99 \times 86$ normally, around the wings and spanwise directions, respectively, where the thickness of the first-layer was $0.001c$. As it moved away from the wing surface in the normal direction, the grid spacing increased by a ratio of about 1.46. The X, Y, and Z-directional dimensions for the background grid were $115 \times 115 \times 74$, respectively (the number of Z-directional points were changed because of the ceiling and ground vicinity). The wing grid's outermost cylindrical boundary had a $2.5c$ radius, with the center point at the wing's spanwise axis. The grid also extended in the spanwise direction from both the wing root and the tip to a $1.5c$ distance. The distances between the background grid boundaries and the wing root in the X and Y directions were $20c$, while in the Z-direction, they varied with the ceiling and ground distances. For the calculations, a 0.02 time step ($\Delta t^*$) was used. Variables such as time step, domain size, first layer thickness and mesh dimensions have already been tested [31], so the suitable values for these variables (stated above) were used directly in this study.

The viscous stress and the pressure at wing surface points were integrated to obtain the aerodynamic forces acting on the wing surface. We define the lift ($L$) on the wing as the component of force perpendicular to the stroke's plane, while the drag ($D$) is the component of force perpendicular to the wing's spanwise axis on the stroke plane. The pressure ($C_p$), lift ($C_L$), and drag ($C_D$) coefficients are equated as $C_p = (p - p_\infty)/(0.5\rho U^2) = p^* - p_\infty/0.5\rho U^2$, $C_L = L/(0.5\rho U^2 S)$ and $C_D = D/(0.5\rho U^2 S)$, respectively, where, $p$ means the pressure, $p_\infty$ means the pressure at infinity, $\rho$ means the fluid's density (air), and $S$ means the wing's area. Additionally, $C_p$ can directly express the pressure difference between a specific grid location and the far field.

The code was tested previously by comparing aerodynamic data obtained with the measured data for a rotating [31] and translating insect wing [40]. It verified that the computed results are in good agreement with experimental measurements. In studying the aerodynamic interactions of the spatially arranged wings [41], the aerodynamic forces of two 3D wings in an in tandem configuration, calculated by our code, also fit well with previous numerical results in Ref. [42]. Additionally, the validation of our method in the presence of wall influence can also be found in our recent work [24], which studied the ground-only effect on a flapping wing. In Ref. [24], the comparisons between the aerodynamic forces at systematic wall clearances calculated by our method, and those experimentally measured by Ref. [23], also demonstrated that our method is reliable and authentic.

### 3. Results and Discussion

*3.1. Combined Ceiling and Ground Effect at Re = 10*

From previous ground-only effect studies, it is known that the force trends are different between $Re = 10$ and $Re = 100$ or higher, and the underlying fluid mechanisms are also different due to the different viscosity effects. Therefore, the combined ceiling and ground effect on aerodynamic forces were first investigated at a very low Reynolds number ($Re = 10$). In the simulations, the combined ceiling and ground effect were examined at $D/c = 1.0, 1.5, 2.0, 2.5, 3.0, 4.0$ and $\infty$.

In the simulations, the transient forces achieved periodicity after three flapping cycles. Thus, the fourth cycle was chosen for analysis and the cycle-averaged lift and drag coefficient ($\overline{C_L}$ and $\overline{C_D}$) were calculated on the fourth cycle. Figure 4 shows the transient aerodynamic lift and drag coefficient ($C_L$ and $C_D$) distributions for varying wall clearances. According to Figure 4, after the periodicity is reached, the transient forces distributions are symmetrical between the upstroke and the downstroke because of the symmetrical wing kinematics. Compared to $D/c = \infty$, the transient lift and drag coefficients at $D/c = 1.0$ exhibit the largest systematic increase for nearly the whole flapping cycle, followed sequentially by $D/c = 1.5, 2.0, 2.5, 3.0$ and 4.0. Then, it is natural that the $\overline{C_L}$ and $\overline{C_D}$ decrease monotonically as $D/c$ increases as shown in Figure 5. At $D/c = 1.0$, the $\overline{C_L}$ and $\overline{C_D}$ increase by 217.35% and 43.89%, respectively; this causes a 120.55% increment in $\overline{C_L}/\overline{C_D}$, indicating that the wing can obtain a significant aerodynamic benefit when flapping in the vicinity

of the ceiling and ground. At $D/c = 2.0$, the $\overline{C_L}$ and $\overline{C_D}$ increase by 81.63% and 11.11%, respectively. When $D/c$ increases to 4.0, the $\overline{C_L}$ can still increase by 41.84%, but the increase in $\overline{C_D}$ is only 5.00%. The lift–drag ratio ($\overline{C_L}/\overline{C_D}$) in Figure 5 also shows the monotonic force trend. These monotonous trends with the changing wall clearance $D/c$ are similar to those in ceiling-only effect in the Ref. [31] and ground-only effect in Ref. [24] at such low *Re*.

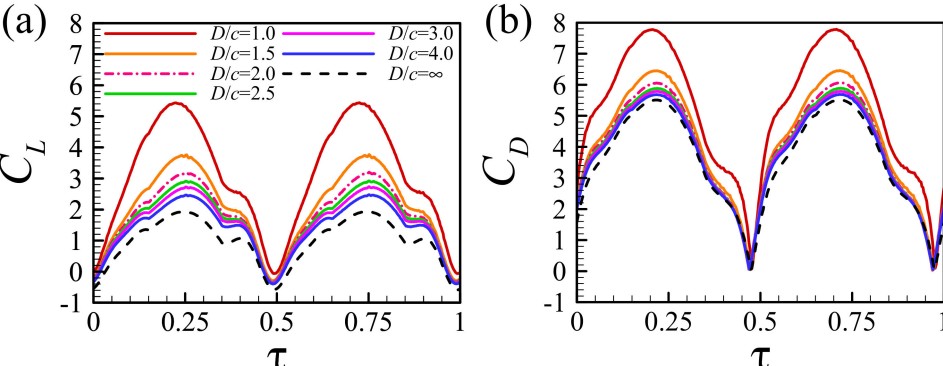

**Figure 4.** The transient aerodynamic (**a**) lift and (**b**) drag coefficient ($C_L$ and $C_D$) distributions for one flapping cycle at systematic wall clearances ($D/c$) and $Re = 10$.

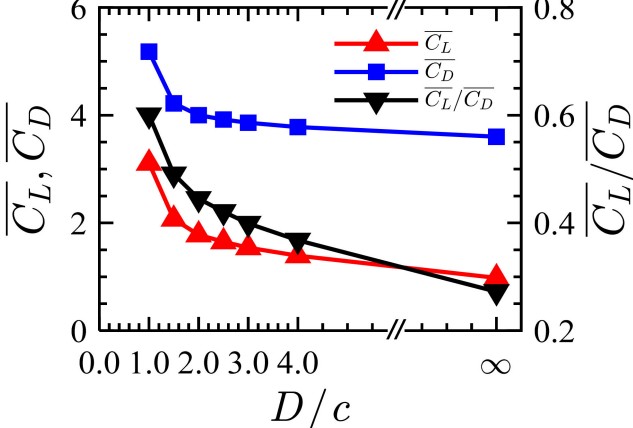

**Figure 5.** Cycled-averaged lift ($\overline{C_L}$) and drag ($\overline{C_D}$) coefficients and lift–drag ratio ($\overline{C_L}/\overline{C_D}$) versus wall clearance ($D/c$) at $Re = 10$.

To know whether or not there exists any aerodynamic interaction (coupling) effect between the ground-only effect and ceiling-only effect, we made more computations for the flapping wing when only a ceiling or only ground exists. The forces of three cases, namely the ground-only case (referred to as *GO*), ceiling-only case (referred to as *CO*), and combined ceiling and ground case (referred to as *CG*), were compared at each of the wall clearance to explore the coupling effect. Figure 6 shows the cycle-averaged lift force enhancement relative to the no-wall case (expressed as $(\overline{C_L} - \overline{C_{L\infty}})/\overline{C_{L\infty}}$) for the above three cases. The sum of the ceiling-only case and the ground-only case (referred to as *CO* + *GO*) is also shown in Figure 6. Table 1 shows the exact values from Figure 6.

From Figure 6, the *GO* case, the *CO* case, and the *CG* case show that the aerodynamic force increases monotonically with decreasing wall clearance. However, the force enhancement in the *CG* case is much larger than that in the *GO* case and the *CO* case when $D/c$ is smaller than or equal to 4.0. From Table 1, it is seen that the increase in the aerodynamic forces for the *CG* case at $D/c = 1.0$ is approximately equal to the sum of the separate effect of the ceiling and ground. This is also true at $D/c = 1.5$ and 2.0.

However, when the wall clearance is larger than 2.0, the combined ceiling and ground effect becomes much larger than the sum of the ceiling-only effect and the ground-only

effect (see Figure 6 and Table 1). For example, at $D/c = 4.0$, the ground-only effect increases $\overline{C_L}$ by 3.06%, which means that the ground effect at this wall clearance is negligible.

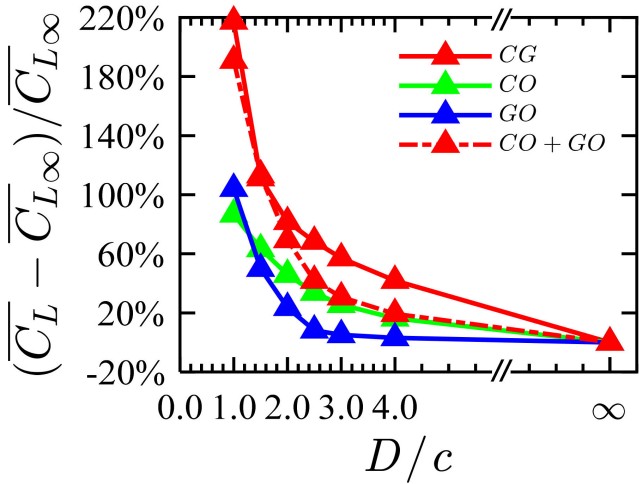

**Figure 6.** The relative increment of cycled-averaged lift coefficient relative to the $D/c = \infty$ case (expressed as $(\overline{C_L} - \overline{C_{L\infty}})/\overline{C_{L\infty}}$) for the combined ceiling and ground case (referred to as $CG$), the ceiling-only case (referred to as $CO$), and the ground-only case (referred to as $GO$) and the sum of the ceiling-only case and the ground-only case (referred to as $CO + GO$) versus $(D/c)$ at $Re = 10$.

**Table 1.** The relative increment of cycled-averaged lift coefficient relative to the $D/c = \infty$ case (expressed as $(\overline{C_L} - \overline{C_{L\infty}})/\overline{C_{L\infty}}$) at $Re = 10$. $CG$, $CO + GO$, $CO$ and $GO$ represent the combined ceiling and ground case, the sum of the ceiling-only case and the ground-only case, the ceiling-only case and the ground-only case, respectively.

| D/c | CG(%) | CO + GO(%) | CO(%) | GO(%) |
|---|---|---|---|---|
| 1.0 | 217.35 | 190.82 | 86.37 | 104.08 |
| 1.5 | 111.22 | 113.27 | 63.27 | 50.00 |
| 2.0 | 81.63 | 69.39 | 45.92 | 23.47 |
| 2.5 | 68.37 | 41.84 | 33.67 | 8.16 |
| 3.0 | 57.14 | 30.61 | 25.51 | 5.10 |
| 4.0 | 41.84 | 19.39 | 16.33 | 3.06 |
| ∞ | 0.00 | 0.00 | 0.00 | 0.00 |

Additionally, the ceiling-only effect increases $\overline{C_L}$ by 16.33% at $D/c = 4.0$. However, for the combined ceiling and ground case at this wall clearance, the $\overline{C_L}$ is increased by 41.84%, which is approximately equal to twice the sum of the separate effect of the ceiling and ground (19.39%). These results reflect that at $D/c = 4.0$, there exists aerodynamic coupling between the ceiling and ground effect.

In the following sections, the underlying fluid physics for the combined ceiling and ground effect were first analyzed, and then the coupling reason at $D/c = 4.0$ was analyzed.

### 3.2. The Reasons for the Combined Ceiling and Ground Effect at Re = 10

We summarize the reasons for the combined ceiling and ground effect as two types. One is called the "downwash-reducing effect", where the wall changes the aerodynamic forces by altering the strength of the downwash associated with the vortex wake; the other is called the "narrow-channel effect", where the fluid dynamics are changed by the channel formed between the wall and the flapping wing, with or without the presence of the vortex wake (downwash).

Let us first see the narrow-channel effect in detail when the ceiling and ground exist together. Figure 7 compares the first stroke's force curves for $D/c$ = 1.0, 1.5, 2.0, 2.5, 3.0, 4.0 and $\infty$. The force enhancement shown in this figure is completely due to the narrow-channel effect, as the wake has not fully developed and there is no chance for the ground and the ceiling to affect the downwash. Compared to $D/c = \infty$, the transient lift coefficient ($C_L$) curve at $D/c$ = 1.0 still exhibits the largest systematic increase for nearly the whole half-flapping cycle, followed sequentially by $D/c$ = 1.5, 2.0, 2.5. However, the transient lift-force coefficient curves at $D/c$ = 3.0 and 4.0 are almost identical to that of $D/c = \infty$. Moreover, the cycle-averaged lift coefficients ($\overline{C_L}$) for the first downstroke at $D/c$ = 3.0 and 4.0 are increased by 7.14% and 3.47%, respectively, compared to $D/c = \infty$. These results indicate that the narrow-channel effect of the combined ceiling and ground effect will not occur when $D/c$ is 4.0 or larger.

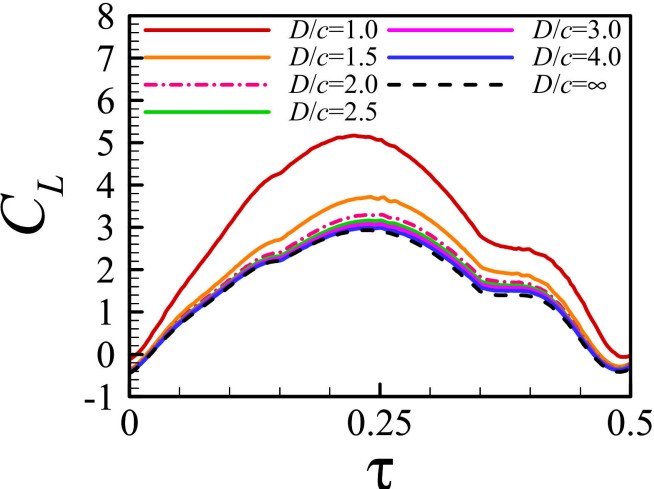

**Figure 7.** The transient aerodynamic lift coefficient ($C_L$) distributions for the first downstroke at systematic wall clearances ($D/c$) and $Re = 10$.

Figure 8 plots the surface pressure ($C_p$) distributions in the middle of the first downstroke ($\tau$ = 0.25) when there is no vortex wake at $D/c$ = 1.0, 4.0 and $\infty$. Observing Figure 8, it is found that the $D/c$ = 1.0 case exhibits both the largest negative pressure zone area on the upper surface and positive pressure zone area on the lower surface among the three cases. For the $D/c$ = 4.0 case, the pressure on both sides is approximately the same as $D/c = \infty$, which is consistent with approximately the same aerodynamic forces shown in Figure 7. Again, this confirms that the narrow-channel effect is insignificant at $D/c$ = 4.0.

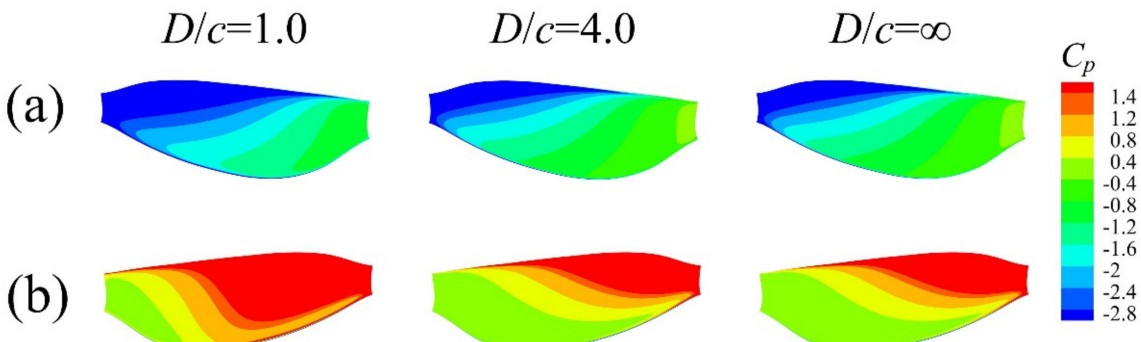

**Figure 8.** Surface pressure contour plots on (**a**) the upper surface and (**b**) the lower surface for $D/c$ = 1.0, 4.0 and $\infty$ in the middle of the first downstroke ($\tau = 0.25$) at $Re = 10$.

Figure 9a,b show the streamlines relative to the wings, along with the incoming flow velocity ($u$) contours on the 2/3*RR* spanwise slice in the middle of the first downstroke ($\tau$ = 0.25), for the $D/c$ = 1.0 and $\infty$ cases ($D/c$ = 4.0 is almost the same as $D/c$ = $\infty$, so is not shown here). Judging from the density of the streamlines and the velocity contours, especially in the squared area, the velocity of the incoming flow of $D/c$ = 1.0 is larger than that of $D/c$ = $\infty$. This is because a narrow channel was formed between the leading edge of the wing and the ceiling. Based on the directions of the velocity vector, the effective angle of attack ($\alpha_e$) of the wing was also calculated and is shown in Figure 9c,d. Comparing the $\alpha_e$ contours for the two cases, it is seen that the effective angle of attack of the $D/c$ = 1.0 is larger than that of $D/c$ = $\infty$. This can easily be seen by comparing the surrounded area of the same contour line with the value of $60^\circ$ in Figure 9c,d. With a larger incoming velocity and effective angle of attack, the wing would produce a larger LEV. This is validated by the fact that the area of negative spanwise vorticity ($\omega_z$) at $D/c$ = 1.0 is larger than that of $D/c$ = $\infty$, as shown in Figure 9e,f. This helps to explain the larger negative pressure on the upper wing surface in Figure 8. The corresponding circulation of the LEV calculated in Figure 10 also shows the consistency, with the circulation along the wingspan for $D/c$ = 1.0 being larger than that of $D/c$ = $\infty$. Here, the increase in the incoming flow velocity and angle of attack in Figure 9 may also be regarded as the result of the existence of an image LEV, as explained in Ref. [31], which studied the ceiling-only effect. The classical image approach was also employed in Ref. [17] to study the ground effect for a fixed-wing. This classical method is based on the linear potential theory. Thus, it should be applied cautiously in the analysis of a wall effect for unsteady motion at a high angle of attack. In addition, the wake vortex, which is hard to convect downstream in hovering flight, is one of the main factors affecting the aerodynamics of a flapping wing. The application of the image method to a flapping wing will be more complex than to a fixed wing, because it is necessary to simulate the image's leading-edge vortex and all the shedding wake vortices at the same time.

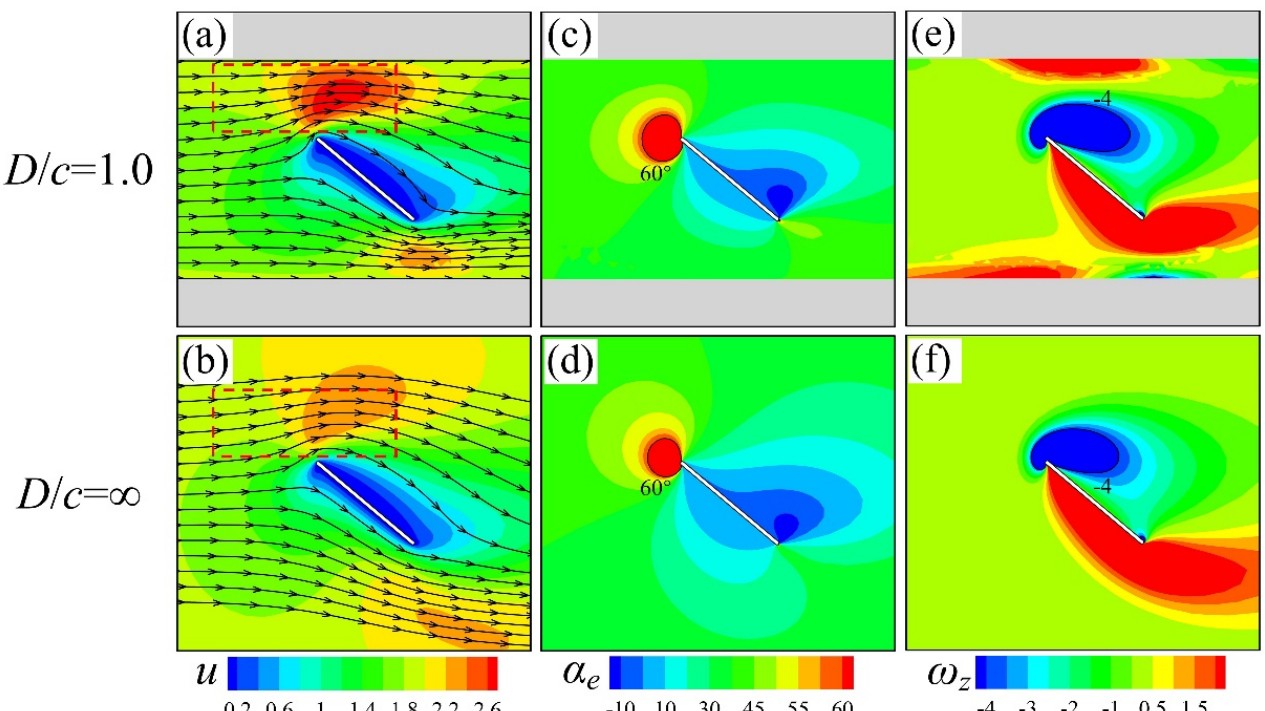

**Figure 9.** The streamlines and velocity ($u$) contours relative to the wings (**a**,**b**), the effective angle of attack ($\alpha_e$) contours (**c**,**d**), and the spanwise vorticity ($\omega_z$) contours (**e**,**f**) in the middle of the first downstroke ($\tau$ = 0.25) at $Re$ = 10. The first row is for $D/c$ = 1.0, and second row is for $D/c$ = $\infty$.

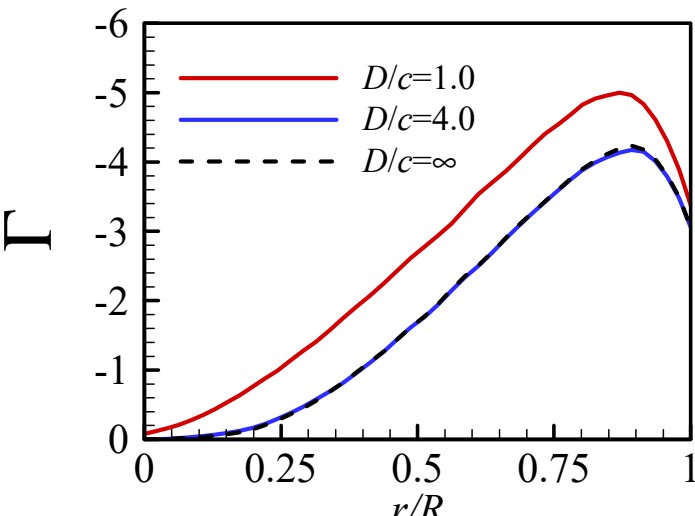

**Figure 10.** The change curves of LEV circulation along the wingspan in the middle of the first downstroke ($\tau = 0.25$) for $D/c = 1.0$, 4.0 and $\infty$ at $Re = 10$.

To explain why there is larger positive pressure on the lower wing surface for the combined ceiling and ground case, the velocity vectors in the inertial frame around the wing section, plus the pressure contours, are plotted in Figure 11. It is found that due to the restriction of the ground at $D/c = 1.0$, a narrow channel is formed between the trailing edge of the wing and the ground, and the velocity vectors are 'squeezed' to align more horizontally than that at $D/c = 4.0$ and $D/c = \infty$. This leads to a larger pressure on the lower surface in Figure 8, which is also evident in the pressure contour distribution and the high-pressure area encircled by the single contour line with a value of 2.0 in Figure 11b. As the clearance increases to $D/c = 4.0$, the air squeezing effect is very weak, the vectors are almost the same between the $D/c = 4.0$ and $\infty$, and finally, the pressure distributions of these two cases are almost the same.

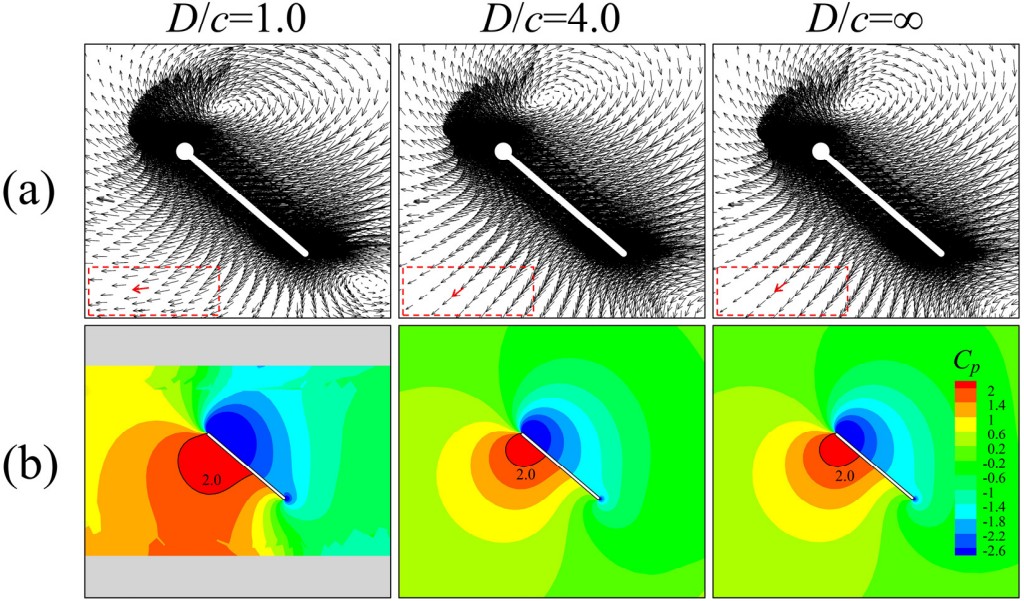

**Figure 11.** (**a**) The velocity vectors; (**b**) the pressure contours on the slice at 2/3 of wingspan in the middle of the first downstroke ($\tau = 0.25$) for $D/c = 1.0$, 4.0 and $\infty$ at $Re = 10$.

From the above analysis of the first stroke when there is no vortex wake, the increase in aerodynamics is only due to the narrow-channel effect. The channel between the wing

and the ceiling increases the relative oncoming flow velocity near the wing's leading edge, and also increases the effective angle of attack. Meanwhile, the narrow channel between the wing and the ground squeezes the fluid and increases the pressure on the lower wing surface.

Referring to Figure 4 again, which shows the force curves after the periodicity has been established, it is noted that the lift coefficient ($C_L$) at $D/c$ = 4.0 becomes larger than that of $D/c$ = ∞. Since the narrow-channel effect is not functioning at $D/c$ = 4.0, the force enhancement at this wall clearance, after the vortex wake is fully developed, can be attributed to the restriction of the wall on the development of the vortex wake and the associated downwash (discussed below). Note that all the results in the remaining section discuss the fourth flapping cycle.

Figure 12 shows the downwash velocity (denoted by $w$) contours when the wing moves from left to right in the top view at $D/c$ = 1.0, $D/c$ = 4.0 and ∞. The dashed boxes are shown to exhibit the regions encountered by the wing at mid-stroke. It is seen that the downwash strength increases as the wall clearance increases. When the wing flaps through this downwash region, smaller downwash enlarges the effective angle of attack, and leads to larger forces.

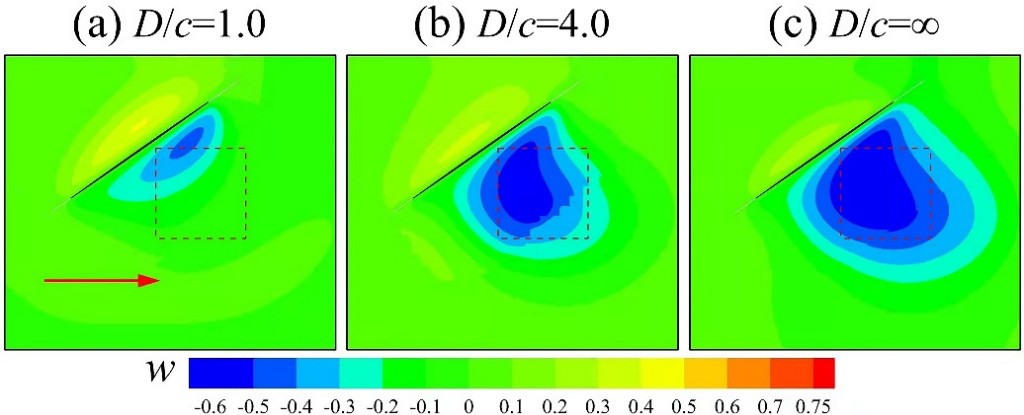

**Figure 12.** The vertical velocity ($w$) contours on the stroke plane (top view) at the beginning of a stroke for (**a**) $D/c$ = 1.0; (**b**) $D/c$ = 4.0; and (**c**) $D/c$ = ∞ at $Re$ = 10. The dashed boxes indicate the regions encountered by the wing at mid-stroke. The wing's stroke direction is given by the red arrow.

Figure 13 shows the fluid features around the wing for $D/c$ = 1.0, 4.0 and ∞ when the wing is in the mid-downstroke ($\tau$ = 0.25). Figure 13a,b exhibit different views of the vortical structures (iso-Q surfaces—green), and the downwash (iso-vertical velocity surface—yellow). The size of the iso-vertical velocity surface on the wing's head indicates the strength of the downwash. For $D/c$ = ∞, because of the high viscosity at such low $Re$ ($Re$ = 10), the trailing-edge vortex cannot separate from the trailing edge of the wing. Rather, it forms an accumulated and thick vorticity layer that remains attached to the wing during a single stroke (marked in Figure 13a,b). When the wing moves back, the thick vorticity layer (or the vortex wake) causes a large downwash at the head of the wing.

At $D/c$ = 4.0, due to the existence of the ceiling and ground, the vortex wake of the previous stroke dissipates quickly. This can be seen by comparing the size of the iso-Q surface of the vortex wake of $D/c$ = 4.0 with that of $D/c$ = ∞ in Figure 13a. Consequently, the downwash caused by the vortex wake is lower near the wing's head. At $D/c$ = 1.0, the vortex wake from the preceding stroke already disappears, so that the downwash is negligible.

Figure 13c gives the spanwise vorticity contour in a spanwise slice, the position of which is shown in Figure 13a. The boxed region in Figure 13c is exaggerated as an inset to show the velocity vectors. It is seen that at low wall clearance, i.e., at $D/c$ = 1.0, there is no vortex wake remaining near the wing head, and the velocity vectors are nearly

horizontal compared to the infinity case ($D/c = \infty$ case). When the wall clearance increases to $D/c = 4.0$, the strength of vortex wake becomes larger and the velocity vectors begin to deflect downwards, meaning that the downwash strength is larger. The vortex wake is the strongest without the wall, and the velocity vectors point entirely downwards, representing the strongest downwash.

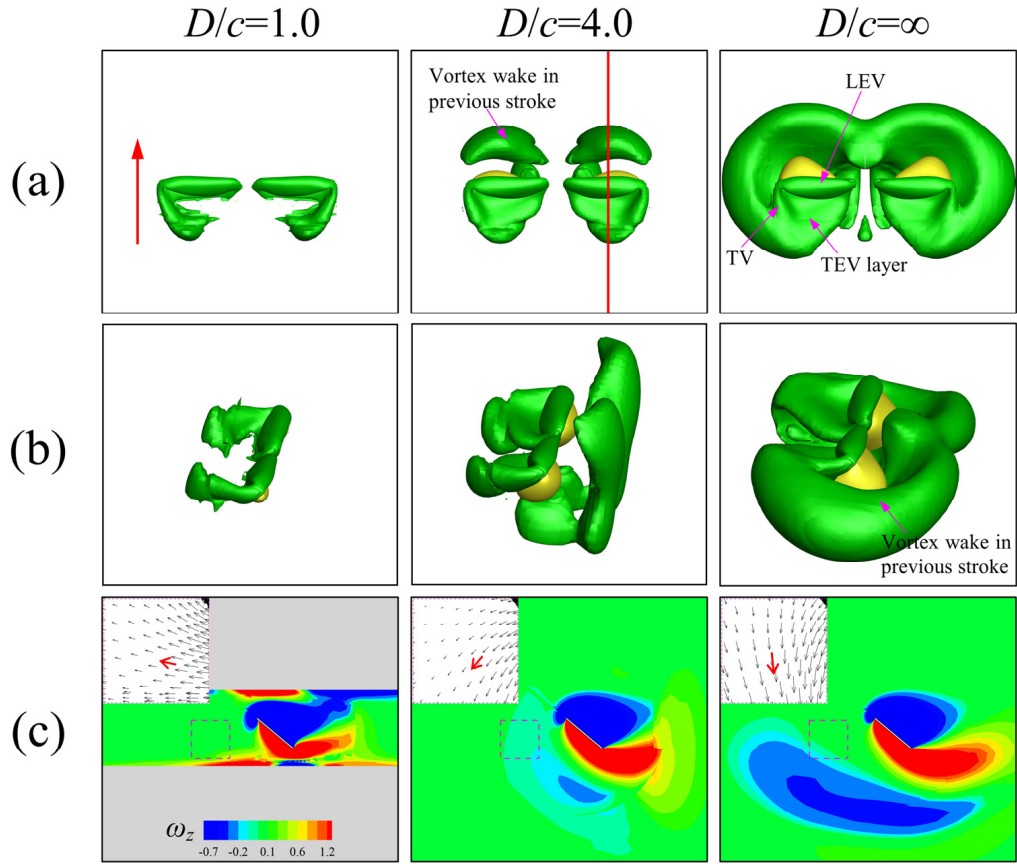

**Figure 13.** (**a**) Top view; (**b**) oblique view of the iso-Q surface (green) and iso-vertical velocity surface (yellow) showing the strength of downwash; and (**c**) spanwise vorticity distributions on a slice at $\tau = 0.25$ for $D/c = 1.0$, 4.0 and $\infty$ at $Re = 10$. The red line in (**a**) gives the slice position. The red arrow in (**a**) indicates the stroke direction.

In short, the combined effect of the ceiling and ground increases the forces through a narrow-channel and downwash-reducing effect. Moreover, it should be noted that when $D/c$ is 4.0 or larger, the force enhancement is mainly due to the downwash-reducing effect.

### 3.3. The Reasons for the Coupling Effect of the Ceiling and Ground at Re = 10

From Figure 6, it is shown that at $D/c = 4.0$, there is a coupling effect when the ceiling and ground exist together, and the force enhancement is much larger than (almost twice) the sum of the ceiling-only effect and ground-only effect. Since the narrow-channel effect disappears at $D/c = 4.0$, we speculate that this coupling effect is achieved through the vortex-wake manipulation.

Figure 14a,b exhibit different views of the vortex structures (iso-Q surfaces—green), and the downwash (iso-vertical velocity surface—yellow) at mid-downstroke ($\tau = 0.25$). It includes the combined ceiling and ground case (referred to as *CG*), the ceiling-only case (referred to as *CO*) and the ground-only case (referred to as *GO*) at $D/c = 4.0$ and infinity case ($D/c = \infty$ case). Observing Figure 14a,b, the vortex structures are almost the same for the ground-only case at $D/c = 4.0$ and the infinity case. There is an accumulated and stable vortex wake ahead of the flapping wing as marked in Figure 14b, which causes the

largest downwash strength. This similar vortex wake, plus the similar aerodynamic forces in Figure 6 and Table 1, indicates that the ground-only effect disappears at $D/c$ = 4.0 or larger. However, for the ceiling-only case at $D/c$ = 4.0, the vortex wake left by the last flapping cycle is reduced in size and is much more scattered and weaker, resulting in a weaker downwash. Hence, the forces of the flapping wing are increased for the ceiling-only case at this wall clearance, as shown in Figure 6 and Table 1. In the case of the combined ceiling and ground, the vortex wake of the preceding stroke is further reduced in size, and is the weakest among the four cases, resulting in the weakest downwash.

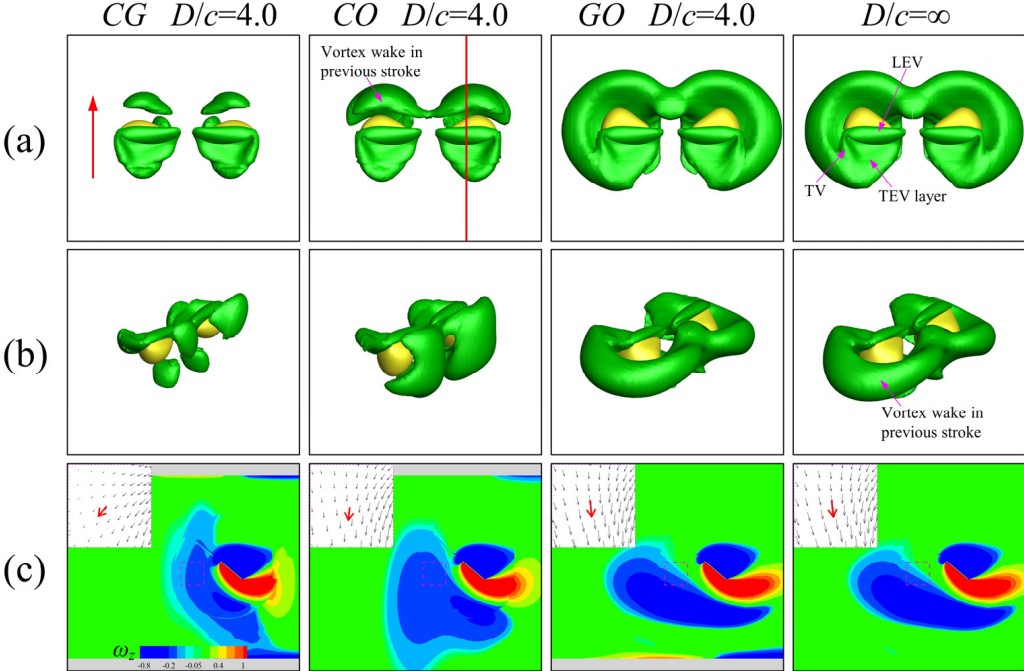

**Figure 14.** (**a**) Top view; (**b**) oblique view of the iso-Q surface (green) and iso-vertical velocity surface (yellow) showing the strength of downwash; and (**c**) spanwise vorticity distributions on a slice at $\tau$ = 0.25 for the combined ceiling and ground case (referred to as $CG$), the ceiling-only case (referred to as $CO$), the ground-only case (referred to as $GO$) at $D/c$ = 4.0 and $D/c$ = ∞ case at $Re$ = 10. The red arrow and red line in (**a**) give the stroke direction and the slice position, respectively.

Figure 14c gives the spanwise vorticity contour of a spanwise slice, the position of which is shown in Figure 14a, along with an inset showing the velocity vectors ahead of the wing. It further shows that the vortex wake and associated downwash for the $GO$ case at $D/c$ = 4.0 is similar to that of the ∞ case, but stronger than that of the $CO$ case. For the $CG$ case, the vortex wake and the downwash are the weakest. In addition, when observing the $GO$ case, the $CO$ case, and the $CG$ case in order, the shape of the vortex wake changes from aligning more horizontally to aligning more vertically. This shape change makes the vortex wake contact the ground in the $CG$ case, which does not happen in the $GO$ case. This phenomenon will be further discussed in the following paragraphs.

From the analysis of the fluid structures in Figure 14, it is seen that at $D/c$ = 4.0, the ground-only effect disappears and the ceiling-only effect reduces the strength of the vortex wake to some extent. However, when the ceiling and ground coexist, the coupling effect on the vortex wake happens, with much quicker vortex-wake dissipation and a more substantial downwash-reducing effect.

To explore how this coupling effect on the vortex wake happens, Figures 15 and 16 plot the evolution of the vortex wake over time from mid-downstroke ($\tau$ = 2/8) to the successive mid-upstroke ($\tau$ = 6/8) for three cases above at $D/c$ = 4.0 (also referred to as $CG$, $CO$ and $GO$). Again, the bigger of the volume of the iso-Q surface, the stronger the vortex wake. Based on the findings of Ref. [43], a vortex ring needs energy input to keep

its form and intensity. The self-induced downwash associated with the vortex ring of the flapping wing can also be regarded as the energy source. The fluid's viscosity and the no-slip boundary on the ceiling could decrease the self-induced downwash above and within the TEV layer. Thus, the input energy of the vortex ring is reduced. Considering that the ground-only effect disappears at $D/c = 4.0$, it is reasonable that the vortex wake of the ceiling-only case would dissipate more rapidly than that of the ground-only case. This can be seen in the top view of the time evolution of the vortex structure in Figure 15b,c. In addition, it shows that the dissipation of the vortex wake for the combined ceiling and ground case is faster than that of the ceiling-only case.

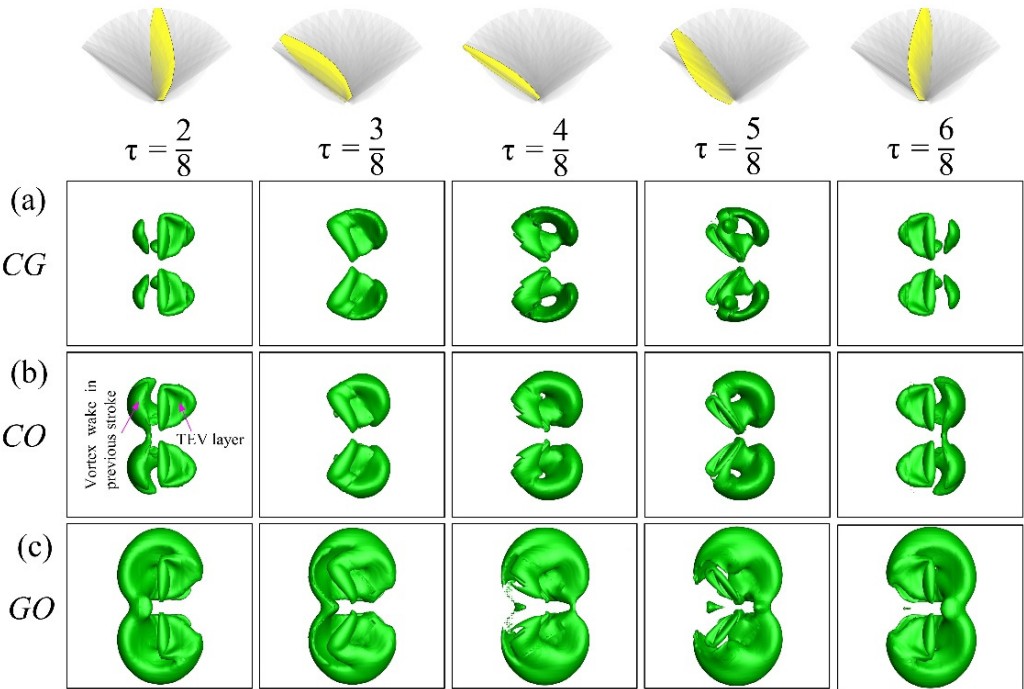

**Figure 15.** Top view of the iso-Q surface (green) with vortex wake for (**a**) the combined ceiling and ground case (referred to as *CG*); (**b**) the ceiling-only case (referred to as *CO*); and (**c**) the ground-only case (referred to as *GO*) at $D/c = 4.0$ for different times at $Re = 10$. The non-transparent wing in the first row indicates the position of the wing at the corresponding time.

The side view of the time evolution of the vortex structures in Figure 16 may explain the reasons for this coupling effect. It can be seen that accompanying the dissipation of the vortex wake, its shape also changes from aligning more horizontally to aligning more vertically from the *GO* case to the *CO* case, then to the *CG* case (see Figure 16, $\tau = 4/8$). This observation is consistent with those in Figure 14c. This elongating in the vertical direction makes the vortex wake nearer to the ground, as shown in Figures 14c and 16a. In turn, the ground, which has no chance to restrict the self-induced downwash below and within the TEV layer at $D/c = 4.0$ in the ground-only case, now has the chance to affect the vortex wake and restrict its downwash. It is believed that these are the reasons for the coupling effect between the ceiling and the ground. Accompanying the further downwash-reducing effect from the ground, the vortex wake loses more energy to keep its status and dissipate much more quickly. This can be seen from the last column for $\tau = 6/8$ in Figures 15 and 16, where most of the vortex wake has dissipated for the combined ceiling and ground case. However, for the ground-only case, the vortex wake still keeps its shape and intensity, while the size of the vortex wake for the ceiling-only case is between that of the two of the other cases.

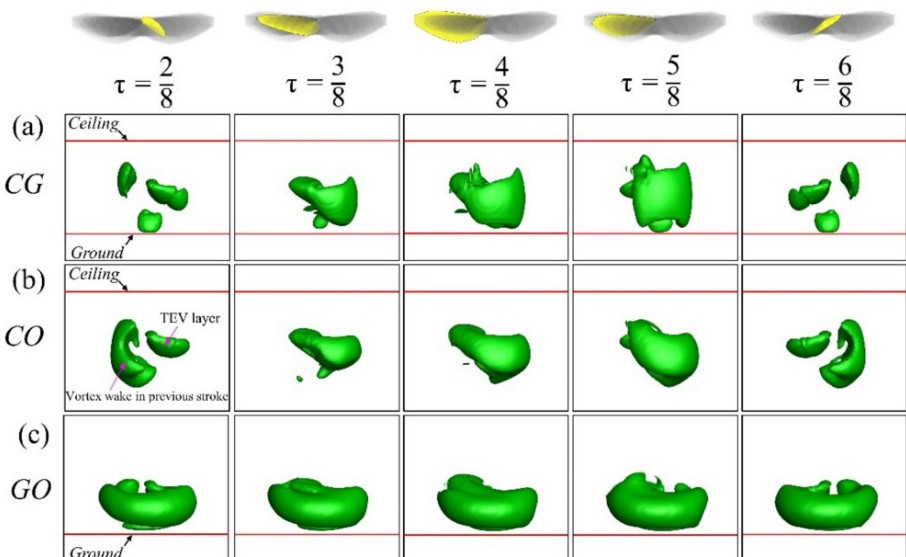

**Figure 16.** Side view of the iso-Q surface (green) with vortex wake for (**a**) the combined ceiling and ground case (referred to as *CG*); (**b**) the ceiling-only case (referred to as *CO*); and (**c**) the ground-only case (referred to as *GO*) at $D/c = 4.0$ for different times at $Re = 10$. The red line means the position of the ceiling and/or the ground. The non-transparent wing in the first row indicates the position of the wing at the corresponding time.

### 3.4. Combined Ceiling and Ground Effect at Re = 100

In the study of the ground effect in Ref. [24], the reasons for $Re = 10$ and $Re = 100$ are different. Thus, it is interesting to see the combined ceiling and ground effect on the aerodynamics of a flapping wing at a higher $Re$. Using the same parameters and kinematics as those at $Re = 10$, we employed the equivalent code to simulate the $Re = 100$ case.

Figure 17 shows the transient aerodynamic lift and drag coefficient ($C_L$ and $C_D$) distributions for varying wall clearances. Compared to $D/c = \infty$, the transient lift and drag coefficients at $D/c = 1.0$ exhibit a large systematic increase for nearly the whole flapping cycle. The $D/c = 1.0$ case is followed sequentially by the $D/c = 1.5$ case. However, the transient lift and drag coefficients decrease rapidly to the lowest values for nearly the whole flapping cycle from $D/c = 1.5$ to $D/c = 2.5$. From $D/c = 2.5$ to $D/c = 4.0$, the transient lift and drag coefficients rise again and almost overlap with the $D/c = \infty$ case. Consequently, it is shown in Figure 18 that the cycle-averaged lift and drag coefficients ($\overline{C_L}$ and $\overline{C_D}$) firstly decrease to the minimum at about $D/c = 2.5$, and then recover as $D/c$ increases further. Thus, the force behavior at $Re = 100$ exhibits a non-monotonous trend of 'three force regimes' as the wall clearance $D/c$ changes. This behavior is different from the monotonous trend at $Re = 10$.

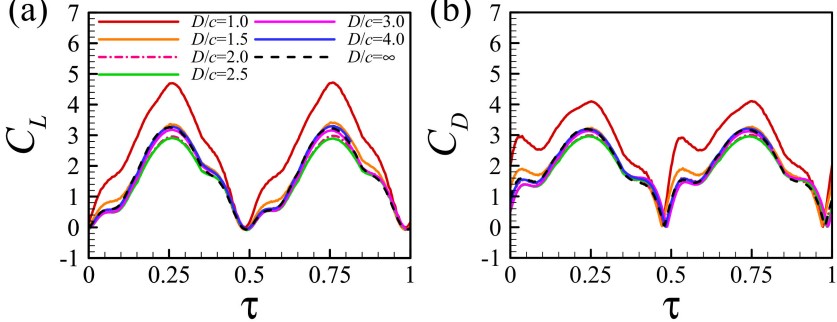

**Figure 17.** The transient aerodynamic (**a**) lift and (**b**) drag coefficient ($C_L$ and $C_D$) distributions for one flapping cycle at systematic wall clearances ($D/c$) and $Re = 100$.

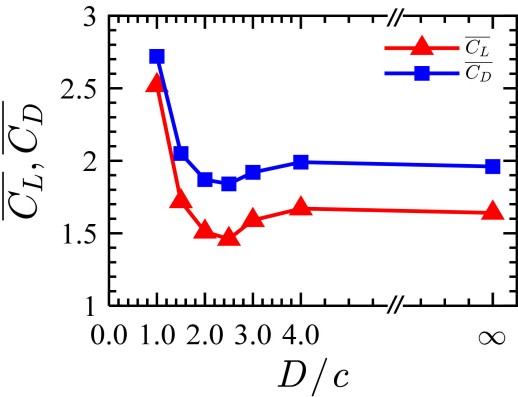

**Figure 18.** Cycled-averaged lift ($\overline{C_L}$) and drag ($\overline{C_D}$) coefficients versus wall clearance ($D/c$) at $Re = 100$.

To know whether or not there exists any aerodynamic interaction (coupling) effect between the ground-only effect and ceiling-only effect at $Re = 100$, we also made more computations for the flapping wing when only the ceiling or only ground exists. Figure 19 shows the force enhancement relative to the no-wall case (expressed as $(\overline{C_L} - \overline{C_{L\infty}})/\overline{C_{L\infty}}$) under the combined ceiling and ground case (referred to as *CG*), the ceiling-only case (referred to as *CO*), the ground-only case (referred to as *GO*), and the sum of the ceiling-only case and the ground-only case (referred to as *CO* + *GO*). The ceiling-only case shows that the aerodynamic force increases monotonically as the wall clearance decreases. However, the ground-only case and the combined ceiling and ground case both show that the aerodynamic forces exhibit a non-monotonous trend of 'three force regimes' as the wall clearance $D/c$ decreases. By comparing the results of the *CG* with those of the *CO* + *GO*, it can be observed that the changes in the aerodynamic force caused by the combined ceiling and ground effect are approximately equal to the sum of the ceiling-only effect and the ground-only effect.

Table 2 gives the exact values from Figure 19. It is shown that the difference between the combined ceiling and ground effect, and the sum of the ceiling-only effect and the ground-only effect, are always smaller than 10% (the maximum difference is 8.54% at $D/c = 1.0$). These results show that the coupling effect of the ceiling and the ground is small at $Re = 100$.

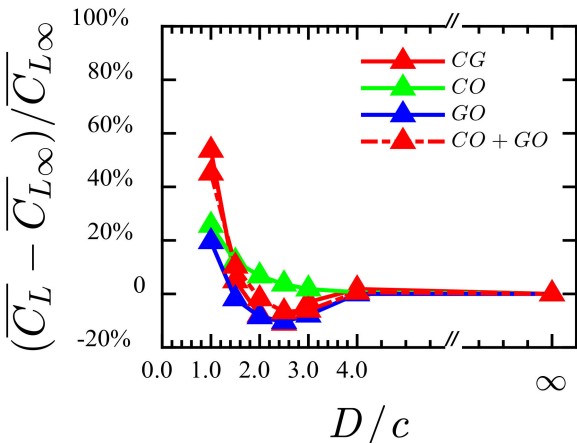

**Figure 19.** The relative increment of cycle-averaged lift coefficient relative to the $D/c = \infty$ case (expressed as $(\overline{C_L} - \overline{C_{L\infty}})/\overline{C_{L\infty}}$) for the combined ceiling and ground case (referred to as *CG*), ceiling-only case (referred to as *CO*), and ground-only case (referred to as *GO*), and the sum of the ceiling-only case and the ground-only case (referred to as *CO* + *GO*) versus wall clearance ($D/c$) at $Re = 100$.

**Table 2.** The relative increment of cycled-averaged lift coefficient relative to the $D/c = \infty$ case (expressed as $(\overline{C_L} - \overline{C_{L\infty}})/\overline{C_{L\infty}}$) at $Re = 100$. $CG$, $CO + GO$, $CO$ and $GO$ represent the combined ceiling and ground case, the sum of the ceiling-only case and the ground-only case, the ceiling-only case, and the ground-only case, respectively.

| $D/c$ | $CG(\%)$ | $CO + GO(\%)$ | $CO(\%)$ | $GO(\%)$ |
|---|---|---|---|---|
| 1.0 | 53.66 | 45.12 | 25.61 | 19.51 |
| 1.5 | 4.88 | 10.37 | 12.20 | −1.83 |
| 2.0 | −7.93 | −1.83 | 6.71 | −8.54 |
| 2.5 | −10.98 | −6.71 | 3.66 | −10.37 |
| 3.0 | −3.05 | −6.10 | 1.83 | −7.93 |
| 4.0 | 1.83 | 0.61 | 0.61 | 0.00 |
| ∞ | 0.00 | 0.00 | 0.00 | 0.00 |

*3.5. The Reasons for the Combined Ceiling and Ground Effect at Re = 100*

To explain the reasons for the non-monotonous aerodynamic force behavior for the combined ceiling and ground case at $Re = 100$, the vortex structures and downwash were also studied. Figure 20 shows the downwash contours when the wing moves from left to right in the top view at $D/c = 1.0$, $D/c = 2.5$ and ∞. It is seen that the downwash strength firstly increases as the wall clearance ($D/c$) changes from 1.0 to 2.5, and then decreases as the wall clearance changes from 2.5 to ∞. When the wing flaps through this downwash region, a smaller downwash enlarges the effective angle of attack, and leads to larger forces. Moreover, the non-monotonous downwash strength causes the non-monotonous aerodynamic force behavior.

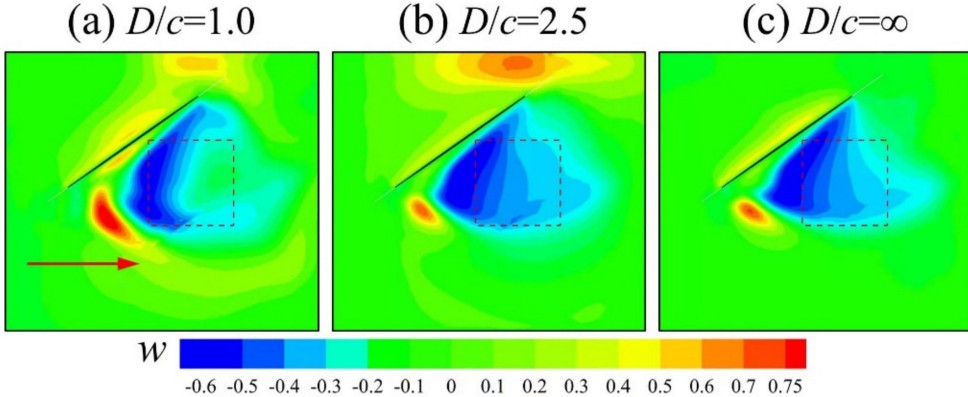

**Figure 20.** The vertical velocity ($w$) contours on the stroke plane (top view) with vortex wake at the beginning of a stroke for (**a**) $D/c = 1.0$; (**b**) $D/c = 2.5$; and (**c**) $D/c = \infty$ at $Re = 100$. The dashed boxes indicate the regions encountered by the wing at mid-stroke. The wing's stroke direction is given by the red arrow.

Figure 21a,b are different views of the vortex structures (iso-Q surfaces) at mid-downstroke ($\tau = 0.25$) for $D/c = 1.0$, 2.5 and ∞. The corresponding iso-vertical velocity surfaces (yellow) on the wing's head are also given. Because of the relatively low viscosity at $Re = 100$, the trailing-edge vortex easily separates from the trailing edge of the wing. Consequently, the LEV, root vortex (RV), TEV, and tip vortex (TV) form the vortex ring for all the three wall clearances. Additionally, the blue dash-dot line and red dashed line indicate the vortex ring formed in this stroke and the vortex wake of the preceding stroke, respectively. For the $D/c = 1.0$ case, the wall clearance is relatively small; thus, the wake is quickly dissipated as it is stretched away from the wing by the walls. Finally, the lowest interaction occurs between the wing and the wake. As the wall clearance increases to

a medium clearance ($D/c$ = 2.5), the vortex wake becomes less stretched and dissipates more slowly; hence, the downwash area is mainly on the wing's path. Thus, it causes more interaction of the wing wake and the forces decrease. However, from $D/c$ = 2.5 to $D/c$ = ∞, due to the low viscosity at $Re$ = 100 and without the bottom wall's restriction, the vortex wake can advect downward and out of the wing's path (see Figure 21b, $D/c$ = ∞). Consequently, the downwash caused by the vortex wake on the wing declines, and the forces recover. The evolution of the vortex wake and the associated downwash is similar to that of the ground-only effect in Ref. [24].

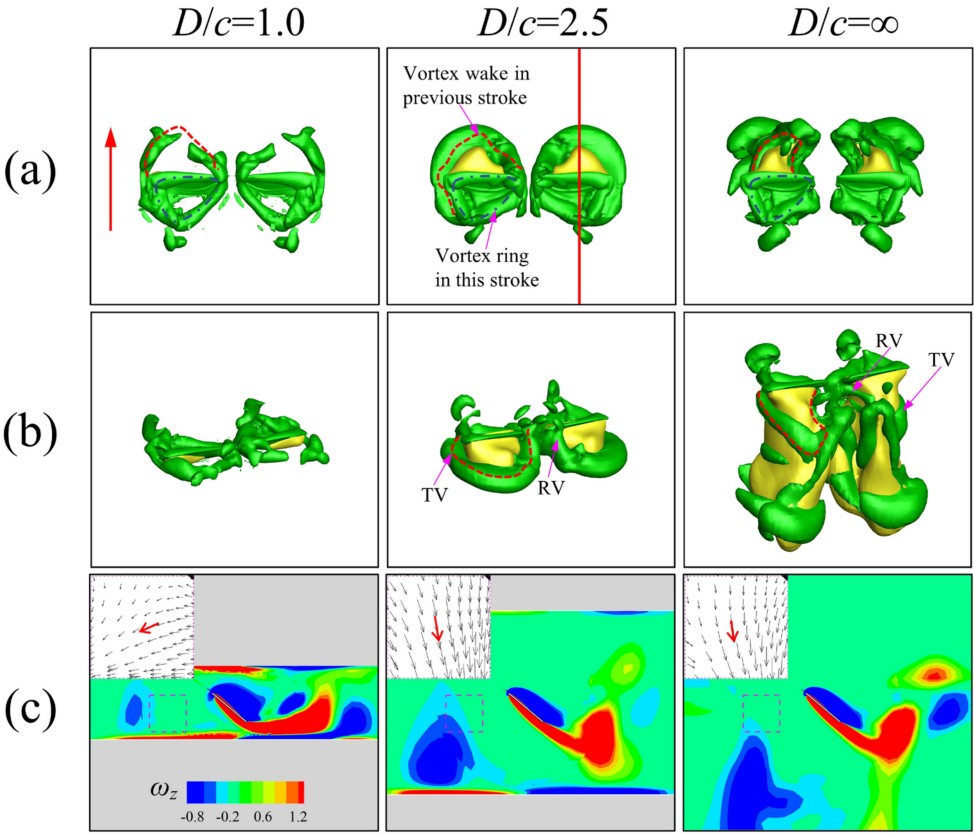

**Figure 21.** (**a**) Top view; (**b**) oblique view of the iso-Q surface (green) and iso-vertical velocity surface (yellow) showing the strength of downwash; and (**c**) spanwise vorticity distributions on a slice at $\tau$ = 0.25 for $D/c$ = 1.0, 2.5 and ∞ at $Re$ = 100. The red arrow and red line in (**a**) give the stroke direction and the slice position, respectively.

Figure 21c shows the spanwise vorticity distribution of a slice pointed in Figure 21a. From $D/c$ = 1.0 to 2.5, the vortex wake changes from being further away to being close in the horizontal direction, and in the meantime, becomes stronger. As $D/c$ further increases to ∞, the wake can convect downwards. Moreover, judging by velocity vectors in the inset of Figure 21c, it is observed that the downward component of the velocity vector first increases and then declines with the increasing wall clearance. Therefore a "three force regimes" behavior is obtained. These findings are similar to Lu et al. [20] and Ref. [24].

As discussed, at $Re$ = 10, the ceiling and ground affect aerodynamic forces in two ways. One is by affecting the vortex wake, and the other is by creating a narrow channel between the edges of the wing and the wall. Here, at $Re$ = 100, we examine the narrow-channel effect from the first stroke. Figure 22 shows the cycle-averaged lift and drag force coefficients ($\overline{C_L}$ and $\overline{C_D}$) for the first stroke. It shows a monotonic trend with decreasing wall clearance. This indicates that the narrow-channel effect in the combined ceiling and ground case always monotonically increases the forces, no matter what the $Re$ is. However, the narrow-channel effect almost disappears at $D/c$ = 2.0, with the cycle-averaged lift coefficient increasing

by only 5%. Recall that at $Re = 10$, the cycle-averaged lift coefficient increment caused by the narrow-channel effect is 15.19% at $D/c = 2.0$, and always larger than 5% unless $D/c$ is equal to or larger than 4.0. Compared to the narrow-channel effect at $Re = 100$ with $Re = 10$, it is concluded that the narrow-channel effect will disappear at a smaller wall clearance for a higher Reynolds number.

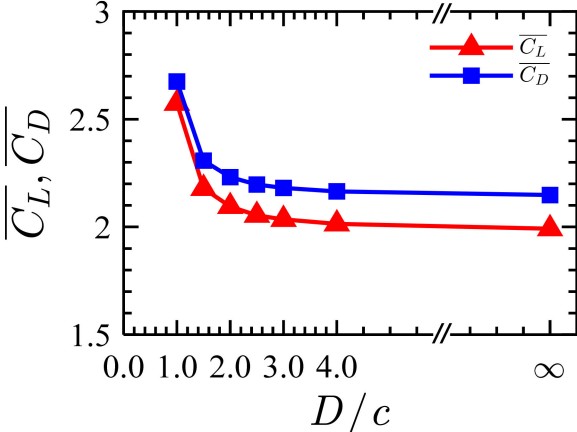

**Figure 22.** Cycled-averaged lift ($\overline{C_L}$) and drag ($\overline{C_D}$) coefficients for the first downstroke as a function of wall clearance ($D/c$) at $Re = 100$.

Figure 23 explains the narrow-channel effect by comparing the pressure ($C_p$) distributions, the effective angle of attack ($\alpha_e$) contours, and velocity ($u$) contours at a 2/3 wingspan slice at the first mid-downstroke for both the $D/c = 1.0$ and $\infty$ case at $Re = 100$. From Figure 23a,b, it is found that the narrow channel between the wing and the ground can produce a larger positive pressure below the wing. Figure 23c–f show that both the magnitude of the incoming flow velocity and the effective angle of attack for $D/c = 1.0$ are larger than that for the infinity case, so that the flapping wing produces a larger LEV and more significant negative pressure on the upper surface of the wing.

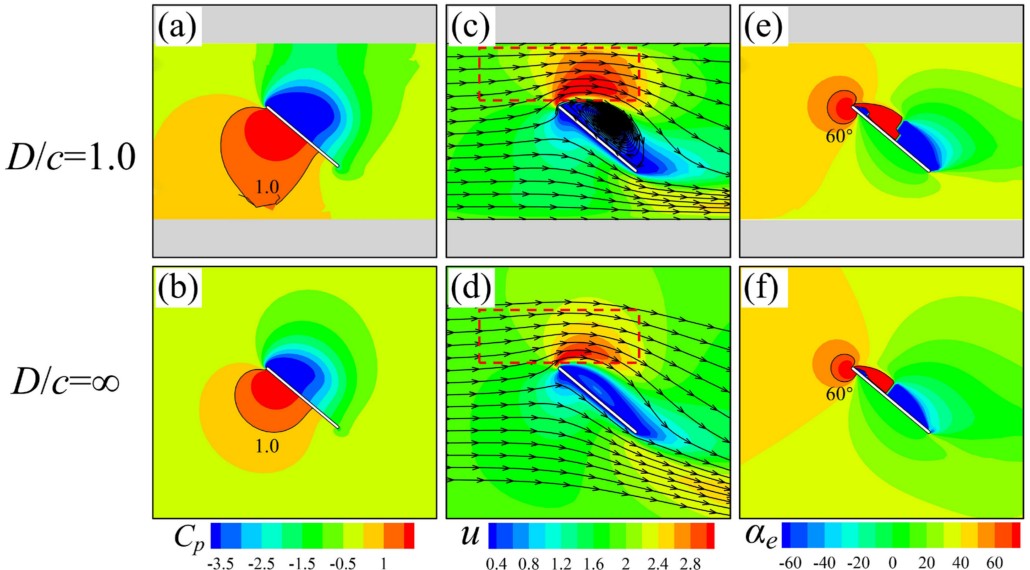

**Figure 23.** The pressure ($C_p$) distributions (**a,b**), the streamlines and velocity ($u$) contours relative to the wing (**c,d**), and the effective angle of attack ($\alpha_e$) contours (**e,f**) at the first mid-downstroke ($\tau = 0.25$) and $Re = 100$. The first row is for $D/c = 1.0$, and the second row is for $D/c = \infty$.

## 4. Conclusions

The combined aerodynamic effects of the ceiling and the ground on normal hovering flapping wings were studied using the computational fluid dynamics method at various wall clearances and at two Reynolds numbers ($Re$ = 10 and 100). The underlying mechanisms of the combined ceiling and ground effect and the coupling effect at $Re$ = 10 are explained.

The combined effect of the ceiling and the ground changes the aerodynamic forces through two effects, namely the narrow-channel effect and the downwash-reducing effect. The narrow-channel effect enhances the suction force on the upper wing surface by increasing the incoming flow velocity and the effective angle of attack. It also produces more significant positive pressure on the lower wing surface by squeezing fluid. The narrow-channel effect decreases monotonically with increasing wall clearance at both $Re$ = 10 and 100. However, at a higher $Re$, it disappears at a smaller wall clearance.

The downwash-reducing effect decreases the strength of the downwash associated with the vortex wake, and consequently, increases the aerodynamic forces. At a low $Re$ ($Re$ = 10), the ceiling and ground decrease the downwash strength monotonically with decreasing wall clearance, by helping the vortex wake to shed from the wing, and meanwhile, dissipate quickly. However, at $Re$ = 100, the downwash strength is changed non-monotonically by the interaction between the vortex wake and the walls. As the wall clearance increases, the distance between the vortex wake and the wing first decreases and then increases, leading to a non-monotonic change in the downwash intensity caused by the vortex wake on the wing. This non-monotonic downwash-reducing effect and the narrow-channel effect eventually lead to the three force regimes (force enhancement, force reduction, and force recovery) at $Re$ = 100.

At $Re$ = 10, there is a coupling effect, where the force enhancement caused by the combined ceiling and ground effect is much larger than (almost twice) the sum of the ceiling-only effect and the ground-only effect. The underlying fluid physics are as follows. When the wall clearance is large enough, the ground-only effect on the vortex wake disappears first. However, at the same wall clearance, the ceiling-only effect still functions. It changes the vortex-wake shape from aligning more horizontally to aligning more vertically, thus bringing the shed vortex wake close enough to the ground, and then the ground's downwash-reducing effect comes into play. Finally, this coupling effect causes the vortex wake for the combined case to dissipate much more quickly than in the ceiling-only case and the ground-only case. Unlike at $Re$ = 10, the coupling effect at $Re$ = 100 is small.

**Author Contributions:** Conceptualization, X.M. and G.C.; methodology, X.M.; Z.C. and A.G.; formal analysis, X.M.; Y.H.; Z.C. and G.C.; data curation, Y.H.; writing—original draft preparation, X.M.; Y.H. and G.C.; writing—review and editing, X.M.; A.G. and G.C.; visualization, Y.H. and Z.C.; supervision, G.C.; funding acquisition, X.M. and G.C. All authors have read and agreed to the published version of the manuscript.

**Funding:** This research was funded by grants from the National Natural Science Foundation of China (Grant Nos. 11872293 and 12172276) and the Natural Science Foundation of Shaanxi Province (Grant No. 2022JC-03).

**Institutional Review Board Statement:** Not applicable.

**Informed Consent Statement:** Not applicable.

**Data Availability Statement:** Please contact the corresponding author for any additional data or models used in this paper.

**Conflicts of Interest:** The authors declare no conflict of interest.

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
