# Peer review of "Aerodynamic Effects of Ceiling and Ground Vicinity on Flapping Wings"

_applsci, doi:10.3390/app12084012_

Round 1
Reviewer 1 Report
Great work overall. Very interesting analysis of the ground and ceiling effects along with their coupling. It would be interesting to see the effect of side walls included in this for the future.
Reviewer 2 Report
In this work the combined ceiling and ground effect on the aerodynamics of a hovering flapping wing (for Micro Air Vehicles) has been investigated numerically with CFD simulations. The research is attractive for aerodynamic and engineering applications. The paper is quite of good quality, but some requests and suggestions have been provided to increase the quality of the work.
The introduction is quite complete. You could add some references on the aerodynamics studies of the front wing of racing cars. In this sector very important are the devices that can be added in it to vary the aerodynamic forces. For example, you can add:
- Basso, M.; Cravero, C.; Marsano, D. “Aerodynamic Effect of the Gurney Flap on the Front Wing of a F1 Car and Flow Interactions with Car Components”. Energies, 2021, Vol. 14, Issue 8, p. 2059.
- Fernandez-Gamiz, U.; Gomez-Mármol, M.; Chacón-Rebollo, T. Computational Modeling of Gurney Flaps and Microtabs by POD Method. Energies 2018, 11, 2091.
These two papers adopt RANS model to analyse the effect of the Gurney flap for car racing application. It can be useful to consider similar devices also for MAV by showing the vortex and fluid structure after this device that work in ground effect and increase the downforce of the vehicle.
The governing equations section is complete both for the wing motion, both for the Navier-Stokes. The geometrical characteristics have been reported dimensionless. The boundary section could be schematized in Figure 3 or in a Table. It is not clear what is the turbulence model adopted. The discretization grid is shown, but you must report the Y+ contour. Then, the mesh sensitivity for the choice of the grid has been made?
The result section is very complete and detailed. You could enlarge some legends (as in Figure 7), to zoom the vectors of Figure 11 and in general to improve the quality of the contours (often the pictures are too small).
The conclusions are supported by the results, but you could better summarize your contribution.
Reviewer 3 Report
see attached document

Reviewer 4 Report
The authors investigate the aerodynamic effects of ceiling and ground’s vicinity on flapping wings. The work is contains numerical calculations without comparison with experiments.
In general the paper is well built, the various sections are well presented, with adequate figures. The topic is interesting and the way the authors present it allows the readers to understand it.
Some minor issues should be improved:
- The dimensions of the flapping wing: please attach an additional drawing with the flapping wing, showing its shape and it relevant dimensions.
- What is the frequency of the flapping ?
- Please provide additional information on the code used to predict the performance of the flapping wing, as well as its accuracy.
- Has the code been verified against other existing methods ? Please discuss.
Round 2
Reviewer 2 Report
All my questions and suggestions have been answered and added in the revised paper. Now it is ready for the publication, being increased its quality.
Reviewer 3 Report
Thanks for the in-depth response to my comments.
I would still like to question the discussion of the frequency effect. The expression of the Strouhal number can take different forms; in essence it is the ratio of the unsteady and steady components of the rate of change of momentum. In the paper, the calculations are stated to represent a frequency of 150 Hz, so with U and c a formulation is possible.
Also, since the production of vorticity scales with the acceleration - all fluid properties remaining constant - the duration of the flapping period does matter.